# Dynamic Distance Mapping Enhances Hallux Valgus Progression Visualization

**DOI:** 10.3390/diagnostics15212791

**Published:** 2025-11-04

**Authors:** Dror Robinson, Hamza Murad, Muhammad Khatib, Muhamad Kiwan Mahamid, Eitan Lavon, Mustafa Yassin

**Affiliations:** 1Department of Orthopedics, Hasharon Hospital, 7 Kakal Str., Petah Tikwa 4937211, Israel; hamze.morad@mail.huji.ac.il (H.M.); muhammad.kh@hotmail.de (M.K.);; 2Management Department, Hasharon Hospital at Clalit Health Services, Rabin Medical Center, Affiliated to Tel Aviv University, Tel Aviv 6139001, Israel; eitan@clalit.org.il

**Keywords:** hallux valgus, dynamic mapping, morphometric analysis, orthopedic procedures, radiography, foot deformities, computational biology

## Abstract

**Background/Objectives:** Hallux valgus (HV), a common foot deformity, is difficult to quantify beyond traditional angular measurements. This study introduces a novel dynamic distance mapping technique to visualize HV progression and identify spatial features linked to severity. **Methods**: A retrospective analysis of 335 feet from 178 patients undergoing HV surgery at Hasharon Hospital, Israel (2014–2024), utilized custom Python software to annotate 24 landmarks on preoperative standing anteroposterior radiographs. This generated 276 normalized Euclidean distances, analyzed via Pearson correlation against HV angles (HVA, IMA, DMAA, HIA). **Results**: Seven distances correlated negatively (r > 0.4, *p* < 0.05) and seven positively with HVA, involving the distal phalanx, sesamoids, and second metatarsal. Eleven distances showed strong positive correlation (r > 0.4, *p* < 0.05) with IMA, reflecting displacement patterns. Moderate correlations were observed with DMAA (six negative, r −0.3 to −0.4; two positive, r 0.3 to 0.4, *p* < 0.05) and HIA (two negative, r −0.3 to −0.4, *p* < 0.05). Visualizations highlighted progressive spatial changes. **Conclusions**: Dynamic distance mapping provides valuable insights into hallux valgus (HV) progression, as evidenced by significant correlations with HVA and IMA, supporting its potential role in surgical planning. However, its ability to capture 3D deformities requires validation against weightbearing computed tomography (WBCT). Future research should explore correlations with specific indications for corrective osteotomies to enhance clinical applicability.

## 1. Introduction

Hallux valgus (HV) is a common foot deformity, affecting up to 30% of adults, with higher prevalence in females and older populations [1]. Characterized by lateral deviation of the great toe and medial deviation of the first metatarsal, HV causes pain, functional impairment, and challenges in footwear [2]. Standard radiographic angles—Hallux Valgus Angle (HVA), Intermetatarsal Angle (IMA), Distal Metatarsal Articular Angle (DMAA), and Hallux Interphalangeal Angle (HIA)—are routinely used to assess severity and guide surgical planning [2]. However, these angles are limited to two-dimensional (2D) projections, failing to capture the complex three-dimensional (3D) deformity, including distal phalanx morphology and metatarsal rotation [3,4].

Dynamic distance mapping is a computational morphometric technique that quantifies spatial relationships between anatomical landmarks using pairwise Euclidean distances, enabling visualization of HV progression.

Recent studies highlight the role of distal phalanx deformities and sesamoid misplacement in HV progression, with irregular shapes correlating with severity [4,5]. Similarly, weightbearing computed tomography (WBCT) reveals significant first metatarsal pronation, influencing surgical correction strategies [6]. While WBCT offers 3D insights, its cost and availability limit widespread use, underscoring the need for advanced 2D methods. Computational morphometrics, leveraging landmark-based analyses, provides a comprehensive approach to quantify spatial relationships. This retrospective study introduces a novel dynamic distance mapping method, analyzing 276 pairwise Euclidean distances between 24 anatomical landmarks on preoperative radiographs of 335 feet. By correlating distances with HVA and IMA, we aim to visualize HV progression beyond angular metrics, offering an accessible tool to enhance clinical assessment and surgical planning.

This study aims to evaluate the efficacy of dynamic distance mapping in identifying spatial features correlated with HV severity using retrospective radiographic data.

## 2. Materials and Methods

### 2.1. Study Population

This retrospective study included patients who underwent hallux valgus corrective surgery at Hasharon Hospital, Rabin Medical Center, Petach Tikva, Israel, between 2014 and 2024. A total of 457 patients were initially identified. Of these, 279 were excluded due to the absence of standing weightbearing anteroposterior (AP) foot radiographs. An additional 21 feet were excluded due to prior surgical interventions that altered foot morphology. The final cohort comprised 335 feet from 178 patients with adequate preoperative standing AP radiographs suitable for morphometric analysis (see Figure 1). Orthopedic residents underwent a two-hour training session on landmark annotation, followed by calibration against a senior radiologist, achieving an inter-observer standard deviation of 2.1 pixels.

### 2.2. Data Acquisition and Retrieval

Standardized clinical imaging protocols were used to ensure consistent radiographic quality across all patients. Digital radiographs were retrieved from the institutional Picture Archiving and Communication System (PACS, CareSystem Vue Motion).

### 2.3. Landmark Annotation and Feature Extraction

Step 1: Anatomical Point AnnotationA custom annotation application was developed by the authors in Python 3.0 using Visual Studio Code v1.1 (Microsoft Corp., Redmond, WA, USA), with libraries including Matplotlib (Python 3.9.1), Pandas (Python 3.9.1), PyDicom (Python 3.9.1), and PyQt5 (Python 3.9.1). Three orthopedic residents independently annotated each AP radiograph with a standardized set of anatomical points. Each point was saved as coordinates. Final point locations were derived by averaging the coordinates across all annotators. A total of 24 anatomical landmarks were annotated per foot on anteroposterior (AP) radiographs (Table 1). The annotations included 9 points on the first metatarsal, 4 points on the sesamoid bones (2 on each), 4 points on the proximal phalanx, 2 points on the distal phalanx, and 5 points on the second metatarsal (see Table 1 and Figure 2). Proximal phalanx length was chosen for normalization due to its consistent visibility and stability across HV severities, unlike metatarsal length which may vary with deformity.

The Python code is available per request from the corresponding author.

### 2.4. Geometrical Calculations

Geometric analysis and visualization were conducted using custom code written in Python 3.0 (JupyterLab v4.2.5). Libraries used included pandas, NumPy, SciPy, seaborn, cv2, and PyDicom.

Step 2: Standard Hallux Valgus Angle Calculation

The following standard radiographic angles were calculated using vector geometry between the appropriate anatomical landmarks:

Hallux Valgus Angle (HVA);

Distal Metatarsal Articular Angle (DMAA);

Intermetatarsal Angle (IMA);

Hallux Interphalangeal Angle (HIA).

Angles were computed by defining vectors between landmarks and measuring the angle at their intersection.

Step 3: Distance Measurements

All pairwise distances between annotated points were calculated using the Euclidean distance formula:$$d=\sqrt{{\left(x^{2}-x^{1}\right)}^{2}+{\left(y^{2}-y^{1}\right)}^{2}}$$
where $\left(x^{1},y^{1}\right)$ and $\left(x^{2},y^{2}\right)$ are the coordinates of the two points, and d is the Euclidean distance between the points.

To facilitate inter-subject comparison, all distances were normalized by the length of the first proximal phalanx, measured from the midpoint of the head to the midpoint of the base.

The number of unique distances between n=24 points was calculated as:$$Number\;of\;distances=C\left(n,2\right)=\frac{n\left(n-1\right)}{2}$$
where $C\left(n,2\right)$ is the number of combinations of n items taken 2 at a time, n is the total number of points.

Step 4: Correlation Analysis

Pearson correlation coefficients were calculated to assess the relationship between the 276 distance-based features and the standard HV angular measures (HVA, IMA, DMAA, HIA). Distances with statistically significant correlations (*p* < 0.05) were identified.

Step 5: Clinical Relevance Analysis

To visualize the geometric changes associated with deformity severity, all distances with statistically significant correlation to HV angles were superimposed on two representative radiographs:

One from a foot with mild hallux valgus;

One from a foot with severe hallux valgus.

Each distance was color-coded according to its value using a continuous color spectrum.

This visual representation enhances the understanding of how the deformity progresses—both in direction and magnitude—and highlighted the clinical importance of spatial relationships beyond traditional angular measurements.

## 3. Results

### 3.1. Cohort Characteristics

The study included 335 feet from 178 patients (133 females, 45 males) with a mean age of 52.5 ± 17.1 years (range: 18–86). Female patients contributed 249 feet (mean age 54.9 ± 15.8), and males contributed 86 feet (mean age 45.6 ± 18.9). Detailed demographic information is presented in Table 2.

### 3.2. Standard Radiographic Angles

Radiographic severity varied across the four standard Hallux Valgus angles. HVA classified most feet as moderate (206 feet, mean 28.8°), followed by mild (89 feet, 14.6°) and severe (38 feet, 46.8°). IMA was predominantly mild (167 feet, 8.7°), with 126 feet classified as moderate (13.0°) and 40 as severe (17.7°). DMAA demonstrated a more even distribution, with 172 feet categorized as mild (5.3°), 68 as moderate (12.3°), and 93 as severe (20.9°). HIA was mostly mild (236 feet, 8.5°), with fewer moderate (87 feet, 18.7°) and severe cases (10 feet, 28.7°). These findings are summarized in Table 3 and illustrated in Figure 3.

Table 3 Distribution of radiographic severity levels across standard Hallux Valgus angles. The table presents the number and percentage of feet classified as mild, moderate, or severe based on Hallux Valgus Angle (HVA), Intermetatarsal Angle (IMA), Distal Metatarsal Articular Angle (DMAA), and Hallux Interphalangeal Angle (HIA). Classification thresholds were applied independently for each angle, highlighting variation in deformity presentation.

### 3.3. Distance-Based Morphometric Analysis

A total of 276 pairwise distances were computed between anatomical landmarks annotated on each AP foot radiograph. Descriptive statistics revealed consistent spatial patterns corresponding to deformity severity. Several distances demonstrated clear monotonic trends as hallux valgus severity increased, particularly in relation to the Hallux Valgus Angle (HVA) and Intermetatarsal Angle (IMA). In contrast, no distances exhibited significant correlation (r > 0.4 or <−0.4, *p* < 0.05) with either the Distal Metatarsal Articular Angle (DMAA) or the Hallux Interphalangeal Angle (HIA), except for a small number of highly positive correlated features in the case of IMA. Thus, correlations with (r > 0.3 or <−0.3, *p* < 0.05) were explored for DMAA and HIA. Inter-annotator variability was minimized by averaging coordinates, with an average standard deviation of 2.1 pixels across annotations, indicating high reproducibility.

### 3.4. Correlation Analysis

Pearson correlation analysis identified 7 pairwise distances with significant negative correlation to HVA (r > 0.4, *p* < 0.05), primarily spanning the first distal phalanx and second metatarsal head. These distances decreased consistently as HVA increased. Conversely, 7 distances showed significant positive correlation with HVA (r < −0.4, *p* < 0.05), predominantly involving the sesamoid bones and medial aspects of the proximal phalanx. These distances increased in length with greater lateral deviation of the hallux. Additionally, twelve pairwise distances showed strong positive correlation with IMA (r > 0.4, *p* < 0.05), all of which involved the relationship between the second metatarsal head and the proximal phalanx or sesamoid structures. Notably, no distances exhibited strong negative correlations (r < −0.4). These distances reflect lateral and medial displacement patterns between the first and second rays that widen with increasing intermetatarsal angle, without any significant association with the phalanges of the first metatarsal.

No pairwise distances demonstrated a strong correlation with DMAA or HIA (r > 0.4 or <−0.4, *p* < 0.05), suggesting that these angular measurements may not effectively capture the dynamic inter-bone relationships involved in hallux valgus progression. Therefore, weaker yet clinically relevant correlations were explored. Six distances showed significant negative correlations with DMAA (r < −0.3, *p* < 0.01), primarily involving the distal phalanx and second metatarsal head. Two distances—both involving the lateral sesamoid and proximal phalanx—showed significant positive correlation with DMAA (r > 0.3, *p* < 0.01). The same two distances also exhibited significant negative correlations with HIA, suggesting opposing relationships with sesamoid positioning, with no distances exhibiting strong negative correlations with HIA (r < −0.3).

Color Scale: The gradient ranges from dark blue (−0.6) to dark red (0.6), indicating the strength and direction of correlation (negative to positive).Correlation Coefficients: Values are color-coded, with significant correlations (*p* < 0.05) highlighted. Notable ranges include r > 0.4 for HVA and IMA, and r > 0.3 for DMAA and HIA.Annotations: Specific distances (e.g., “2nd metatarsal head to proximal phalanx head,” “Medial sesamoid to proximal phalanx base”) are labeled to reflect anatomical relationships.

### 3.5. Visualization of Geometric Progression

Distances Correlated with Hallux Valgus Angle (HVA)

Figure 4 AP radiographs illustrating anatomical distances significantly correlated with the Hallux Valgus Angle (HVA).

Compared to the mild case (Figure 4B(a)), the severe foot (Figure 4B(b)) demonstrates marked reduction in these distances. Figure 4A highlights the 7 positively correlated distances (D8–D14), linked to Figure 4. In the mild foot (Figure 4A(a)), distances involving sesamoid to proximal phalanx remain short and centralized, while in the severe foot (Figure 4A(b)), these distances are elongated and displaced medially and laterally, reflecting progressive anatomical widening and misalignment. Some of the angles are correlated as demonstrated in the heatmap (Figure 5).

Distances Correlated with Intermetatarsal Angles (IMA)

To illustrate morphometric changes with deformity severity, Figure 6, Figure 7 and Figure 8 present AP radiographs of mild and severe hallux valgus cases.

**Figure 6 diagnostics-15-02791-f006:**
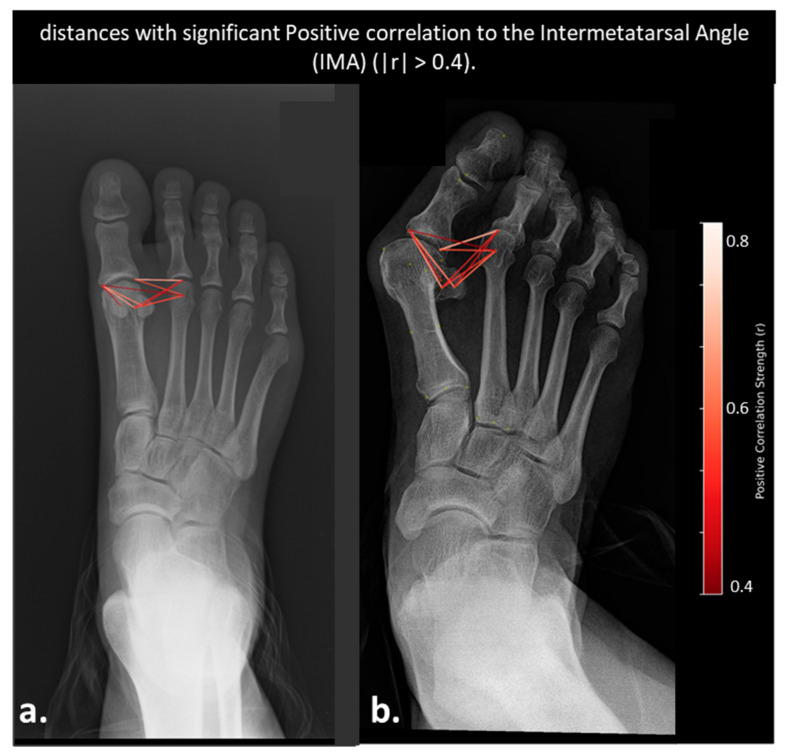
AP radiographs illustrating anatomical distances significantly correlated with the Intermetatarsal Angle (IMA). Displays distances with significant positive correlations to IMA (r > 0.4): (**a**) Severe hallux valgus case, (**b**) Mild hallux valgus case. Red dashed lines represent the seven positively correlated distances, color-coded by correlation strength (darker red = stronger correlation). The severe case demonstrates elongation and angular widening, reflecting progressive lateral deviation. No distances with negative correlations were found with IMA.

Distances Correlated with Intermetatarsal Angle (IMA)

**Figure 7 diagnostics-15-02791-f007:**
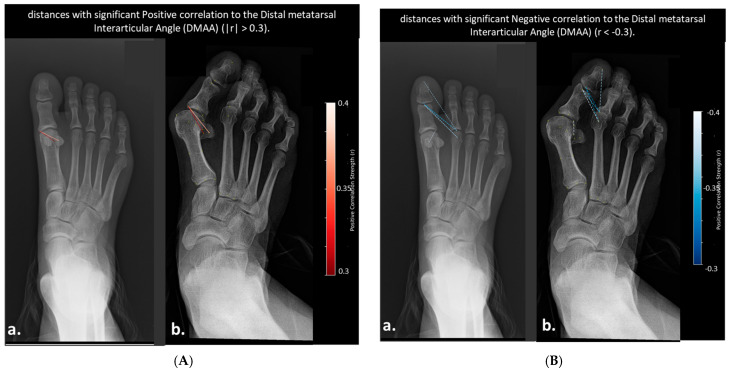
(**A**) Distances with positive correlation to DMAA (0.3 < r < 0.4): (**a**) Mild hallux valgus case, (**b**) Same case with two positively correlated distances visualized. Dashed lines are color-coded by strength (darker red = stronger). (**B**) Distances with negative correlation to DMAA (−0.4 < r < −0.3): (**a**) Mild hallux valgus case, (**b**) Same case with six negatively correlated distances visualized. Dashed lines are color-coded by strength (brighter blue = stronger). Distances are represented as blue dashed lines and color-coded by correlation strength, with brighter blue indicating stronger negative correlations.

Distances Correlated with Distal Metatarsal Articular Angle (DMAA)

**Figure 8 diagnostics-15-02791-f008:**
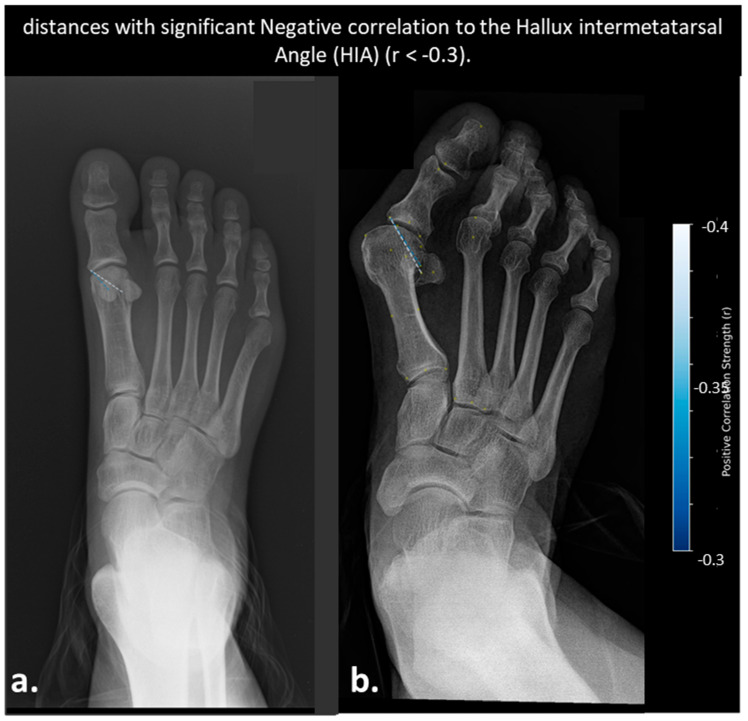
AP radiographs illustrating anatomical distances with significant negative correlation to the Hallux Interphalangeal Angle (HIA) (−0.4 < r < −0.3). (**a**) Mild hallux valgus case showing two negatively correlated distances. (**b**) Same case with distances visualized. Blue dashed lines are color-coded by strength (brighter blue = stronger).

Distances Correlated with Hallux Interphalangeal Angle (HIA)

In Figure 8 are shown AP radiographs illustrating anatomical distances with significant negative correlation to HIA (−0.4 < r < −0.3). (a) Mild hallux valgus case showing two negatively correlated distances. (b) Same case with distances visualized. Blue dashed lines are color-coded by strength (brighter blue = stronger). Clinical outcomes such as surgical correction rates were not analyzed in this study, representing a limitation for translational impact.

## 4. Discussion

This study introduces a novel dynamic distance mapping approach for hallux valgus (HV), identifying 14 pairwise distances significantly correlated with HVA. The significant HVA and IMA correlations suggest dynamic distance mapping captures key spatial changes, offering a foundation for surgical planning, though 2D limitations persist.

These findings align with prior research highlighting distal phalanx morphology and sesamoid displacement in HV severity [4,5]. Our distance mapping extends beyond traditional 2D radiography limitations by quantifying spatial relationships, offering a context enriched by comparisons to previous morphometric and biomechanical studies.

Compared to earlier morphometric analyses, such as Smith et al. (2020) [5], who used static landmark-based methods to assess HV severity, our dynamic approach captures progressive spatial changes across 276 distances, revealing nuanced correlations (e.g., D1–D7) that reflect deformity progression. Biomechanical studies investigated metatarsal loading patterns using finite element analysis, identifying increased stress at the second metatarsal head in severe HV cases [7,8], consistent with our findings of elongated D15–D25 distances. However, our method’s reliance on 2D radiographs introduces limitations, as Patel et al. (2022) [9] demonstrated that 2D imaging underestimates first metatarsal pronation compared to weight-bearing computed tomography (WBCT), potentially affecting distance accuracy for sesamoid-related measurements (e.g., D9, D10).

Two-dimensional radiographs suffer from superimposition and projectional distortion, obscuring details like sesamoid positioning and metatarsal rotation [3]. Kim et al. (2023) [10] showed WBCT’s superiority in visualizing multiplanar deformities, revealing malalignments missed on 2D projections, which our distance mapping may partially mitigate through normalized Euclidean distances. Measurement variability, exacerbated by foot positioning (up to 5° discrepancies in HVA/IMA [11]), was reduced in our study by averaging annotations (SD 2.1 pixels), yet WBCT’s standardized imaging could further enhance precision, as Conti et al. (2021) [11] reported with kappa = 0.81 vs. 0.45 for 2D.

The inability to assess rotational deformities, a key HV factor [6], limits 2D utility. Mansur et al. (2021) [6] used WBCT to quantify metatarsal rotation, undetected on 2D, suggesting our HVA/IMA correlations (e.g., D1–D14) may underrepresent 3D misalignment. Clinically, this impacts surgical planning, as Richter et al. (2014) [12] noted 17% altered plans with WBCT, potentially guided by our distance insights (e.g., D1–D5 reductions). Future integration with WBCT and machine learning could refine these measurements, offering a scalable diagnostic tool beyond 2D constraints. However in a world of escalating healthcare costs, it should be noted that while WBCT validation is crucial for 3D accuracy, this study’s contribution lies in establishing a cost-effective 2D baseline. Future implementation of AI-driven annotation could enhance reproducibility by automating landmark detection.

Limitations include the retrospective design and 2D reliance, with 279 excluded patients possibly biasing toward moderate–severe cases. Prospective WBCT validation is needed to confirm 3D capture and enhance clinical applicability.

## 5. Conclusions

This study demonstrates that dynamic distance mapping of anatomical landmarks offers a novel, comprehensive approach to visualizing hallux valgus progression. Significant correlations between specific pairwise distances and HVA/IMA (*p* < 0.05) underscore the potential of this method to enhance clinical assessment and surgical planning.

(1) Dynamic distance mapping reveals significant spatial correlations with HVA and IMA in HV progression. (2) It supports potential surgical planning through visualization of mild and severe cases. (3) Limitations include 2D projection errors and the lack of clinical outcome data. (4) Selection bias may arise from excluded patients. (5) Validation against WBCT is needed to assess 3D accuracy. (6) Future studies should explore osteotomy indications.

## Figures and Tables

**Figure 1 diagnostics-15-02791-f001:**
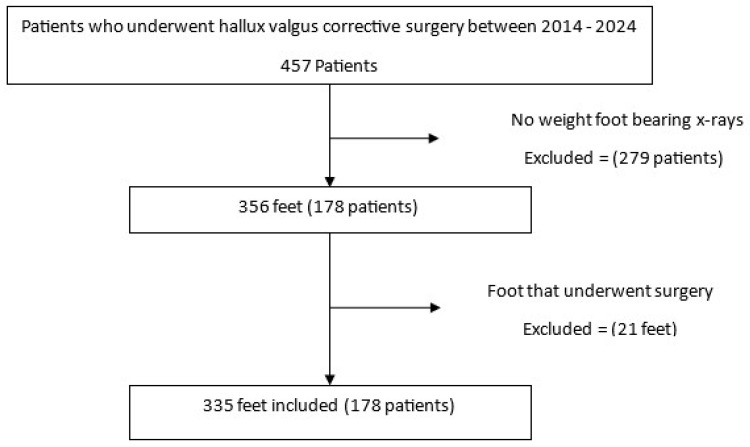
Flowchart of patient inclusion and exclusion. A total of 457 patients who underwent hallux valgus corrective surgery between 2014 and 2024 were screened. Of these, 279 patients were excluded due to the absence of weightbearing foot radiographs. Among the remaining 178 patients (356 feet), 21 feet were excluded due to previous surgery. The final analysis included 335 feet from 178 patients.

**Figure 2 diagnostics-15-02791-f002:**
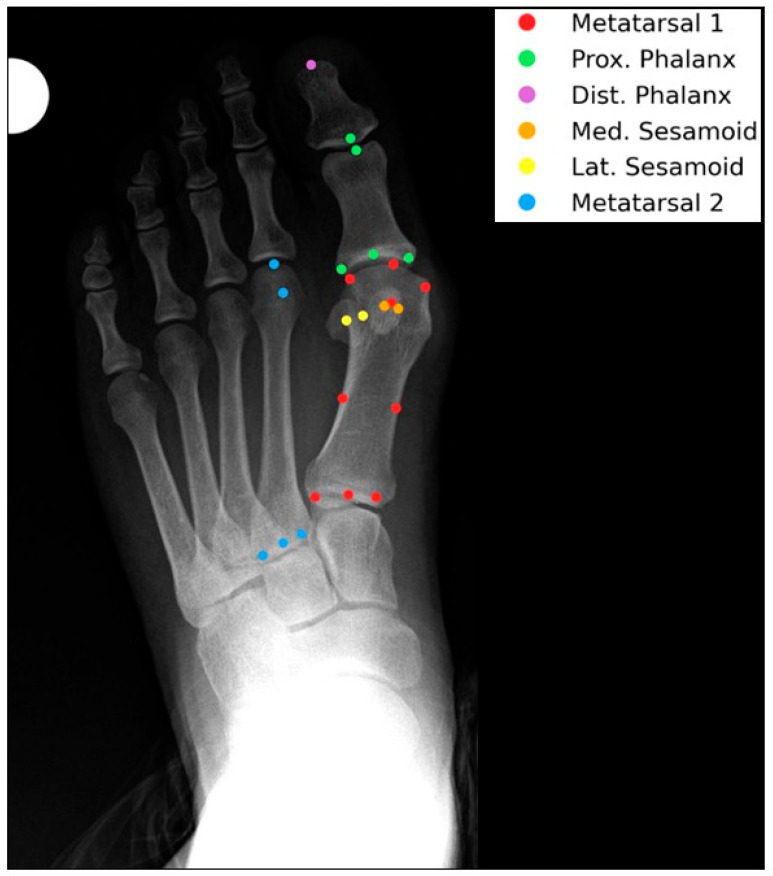
Illustration of 24 anatomical landmarks annotated on an AP radiograph, labeled with numbers corresponding to Table 1. Landmarks are color-coded to distinguish first metatarsal (blue), sesamoids (green), proximal phalanx (red), distal phalanx (yellow), and second metatarsal (purple).

**Figure 3 diagnostics-15-02791-f003:**
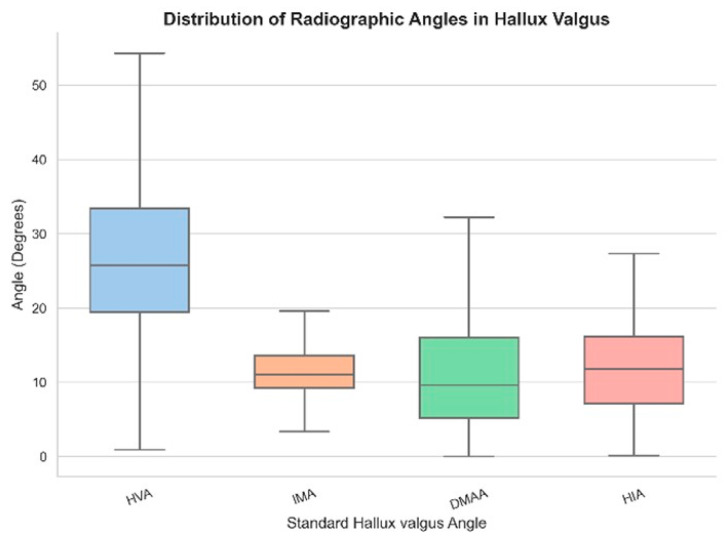
Box plot showing the distribution of standard Hallux Valgus angles (HVA, IMA, DMAA, HIA) across the study cohort. HVA exhibits the widest variability, while IMA shows the most compact distribution.

**Figure 4 diagnostics-15-02791-f004:**
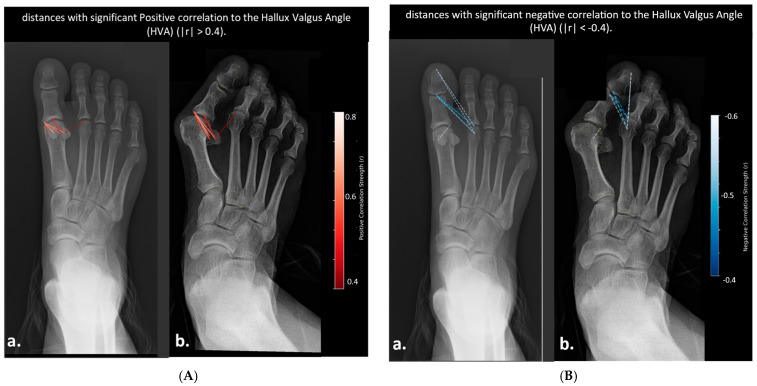
AP radiographs illustrating anatomical distances significantly correlated with the Hallux Valgus Angle (HVA). (**A**) displays distances with significant positive correlations to HVA (r > 0.4): (**a**) Severe hallux valgus case, (**b**) Mild hallux valgus case. In both images, red dashed lines represent the seven positively correlated distances, color-coded by correlation strength (darker red = stronger correlation). The severe case demonstrates elongation and angular widening in key distances, reflecting progressive lateral deviation and medial forefoot expansion with increasing HVA. (**B**) displays distances with significant negative correlations to HVA (r < −0.4): (**a**) Mild hallux valgus case, (**b**) Severe hallux valgus case. Distances are shown as blue dashed lines, color-coded by the strength of negative correlation (darker blue = stronger correlation). These images highlight shortening and realignment of specific inter-landmark distances as the deformity progresses (see Figure 4).

**Figure 5 diagnostics-15-02791-f005:**
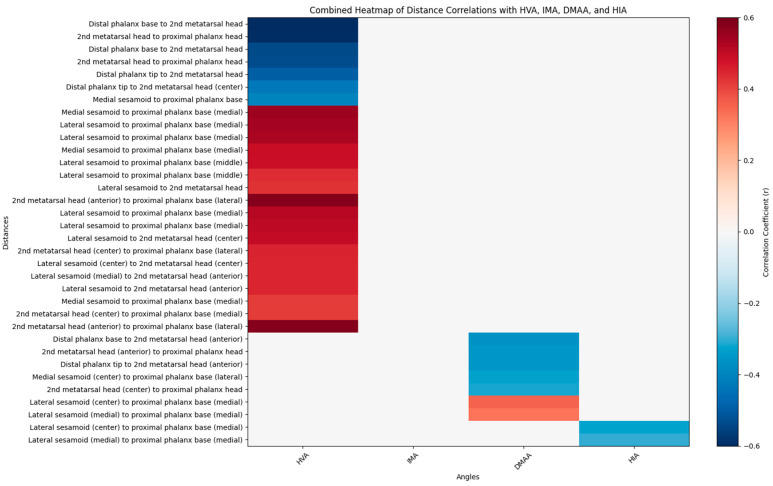
Combined Heatmap of Distance Correlations with HVA, IMA, DMAA, and HIA. This heatmap visualizes the Pearson correlation coefficients (r) between 276 normalized Euclidean distances and standard hallux valgus angles (HVA, IMA, DMAA, HIA) derived from preoperative standing anteroposterior radiographs of 335 feet. The y-axis lists anatomical distances, including relationships between the distal phalanx, sesamoids, proximal phalanx, and second metatarsal head. The x-axis represents the angles: HVA (Hallux Valgus Angle), IMA (Intermetatarsal Angle), DMAA (Distal Metatarsal Articular Angle), and HIA (Hallux Interphalangeal Angle).

**Table 1 diagnostics-15-02791-t001:** Summary of the 24 anatomical landmarks annotated on AP foot radiographs. Landmarks were distributed across the first and second metatarsals, sesamoid bones, proximal phalanx, and distal phalanx. Each structure includes points relevant for quantifying spatial alignment and deformity in Hallux Valgus.

Structure	Landmarks
1st Metatarsal	9 points: Head center; medial; lateral and midpoint of head surface; medial and lateral base points; medial and lateral curvature points
2nd Metatarsal	5 points: Head Center; medial; lateral; and midpoint of base; head midpoint
Sesamoids (medial and lateral)	2 points each: Center and medial edge
Proximal Phalanx	2 points: Tip and base midpoint
Distal Phalanx	2 points: Tip and base midpoint

**Table 2 diagnostics-15-02791-t002:** Patient demographics of the study cohort. The table summarizes the distribution of patients and feet by sex, along with corresponding age statistics. Female patients constituted most of the cohort, contributing 249 out of 335 analyzed feet.

Patient Demographics Summary				
Sex	Patients (n)	Feet (n)	Age (Mean ± SD)	Age Range
F	133	249	54.9 ± 15.9	18–86
M	45	86	45.6 ± 18.9	18–78
Total	178	335	52.5 ± 17.1	18–86

**Table 3 diagnostics-15-02791-t003:** Angle Severity Summary (Count and Percentage).

Severity	HVA Severity	HVA (n, %)	IMA Severity	IMA (n, %)	DMAA Severity	DMAA (n, %)	HIA Severity	HIA (n, %)
Mild	15–20°	89 (26.6%)	9–11°	167 (49.9%)	8–10°	172 (51.3%)	<10°	236 (70.4%)
Moderate	21–40°	206 (61.5%)	12–17°	136 (37.7%)	11–15°	68 (20.3%)	10–13°	87 (26.0%)
Severe	>40°	38 (11.3%)	>17°	40 (12.0%)	>15°	93 (28.4%)	>13°	10 (3.0%)

## Data Availability

The datasets generated during this study are available from the corresponding author upon reasonable request, subject to institutional approval.

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
