# Peer review of "Dynamic Distance Mapping Enhances Hallux Valgus Progression Visualization"

_diagnostics, 2025, doi:10.3390/diagnostics15212791_

Round 1
Reviewer 1 Report
Comments and Suggestions for Authors
A few changes are suggested to make the manuscript even more powerful and clear. To make the title more concise and powerful, it should start with the presentation. The abstract, on the other hand, should be shorter by summarizing technical aspects more clearly to fit conventional word restrictions. We recommend aligning the keywords with the MeSH database to make the content more visible. In the results section, you might want to combine Tables 4 and 5 to give a more complete picture of the relationships. Also, you should check that the numbering and cross-referencing for Figures 4 and 5 are the same in the text and captions. The conversation may be much improved by a more in-depth comparison of the specific distance mapping results with previous morphometric or biomechanical investigations. This would give the conversation a context that goes beyond the well-known problems with 2D radiography. Lastly, a small change is needed to fix the difference between the "DMMA" and "DMAA" abbreviations. Addressing these issues will make your great job even better.
Author Response
please see detailed response in file

Reviewer 2 Report
Comments and Suggestions for Authors
Congrats for this retrospective study aimed to develop a new dynamic distance mapping approach to visualize HV progression and identify spatial characteristics correlated with the severity of the deformity.
In the future, it may be useful to correlate the effectiveness of 3D exploration with specific indications for corrective osteotomies.
Author Response
please see attached file with detailed response

Reviewer 3 Report
Comments and Suggestions for Authors
Dear Author,
Thank you for the opportunity to review this article.
Introduction: The introduction provides a solid background on hallux valgus (HV), summarizing its prevalence, standard angular measurements, and the limitations of 2D radiographs. The rationale for developing a dynamic distance mapping method is clear. Though, the definition of “dynamic distance mapping” could be explained more clearly for readers unfamiliar with computational morphometrics. The aim should be stated in a separate paragraph.
Materials: The normalization method (using proximal phalanx length) is reasonable, but justification for why this measure was chosen instead of metatarsal length is lacking. Landmark annotation was performed by orthopedic residents, but their training and calibration are not described. This could affect reproducibility.
Results: The results are comprehensive, with clear tables and visualizations. Strong correlations between specific distances and HVA/IMA are reported. No analysis of clinical outcomes (e.g., surgical correction, recurrence rates) is included, which limits the translational impact.
Discussion: The discussion repeats large portions of the results instead of synthesizing key insights. While the authors highlight limitations, the explanation of why WBCT validation is necessary dominates the section, overshadowing the actual contributions of their method. The possibility of automated landmark detection (AI-driven annotation) is not discussed, though it would directly address reproducibility issues.
Conclusions: The conclusion is too general and does not reflect the specific limitations (2D projection errors, selection bias, lack of outcome correlation).It states that the method is “equivalent to advanced imaging modalities” pending validation, but this is overstated. At best, the study shows potential rather than equivalence. It should only comprise 3-7 clear affirmations based on your research.
Author Response
Thank you for your comments. Please see detailed response in attached file.

Round 2
Reviewer 3 Report
Comments and Suggestions for Authors
The authors have made the required modifications